# Nine-Strain Bacterial Synbiotic Improves Crying and Lowers Fecal Calprotectin in Colicky Babies—An Open-Label Randomized Study

**DOI:** 10.3390/microorganisms10020430

**Published:** 2022-02-12

**Authors:** Malgorzata Bernatek, Jacek Piątek, Marcin Pszczola, Hanna Krauss, Janina Antczak, Paweł Maciukajć, Henning Sommermeyer

**Affiliations:** 1Department of Health Sciences, Calisia University, Nowy Swiat 4, 62-800 Kalisz, Poland; drpiatek@o2.pl (M.B.); hjk12@poczta.fm (H.K.); h.sommermeyer@akademiakaliska.edu.pl (H.S.); 2Department of Genetics and Animal Breeding, Faculty of Veterinary Medicine and Animal Science, Poznan University of Life Sciences, Wolynska 33, 60-637 Poznan, Poland; marcin.pszczola@puls.edu.pl; 3State Hospital Jarocin, Szpitalna 1, 63-200 Jarocin, Poland; janina.antczak@gmail.com; 4GP Clinic Brody, Stanisława Staszica 3, 27-230 Brody, Poland; pawel.maciukajc@gmail.com

**Keywords:** colic, colicky babies, crying, dysbiosis, fecal calprotectin, gut inflammation, gut microbiota, infantile colic, simethicone, synbiotic

## Abstract

The aim of this study (ClinicalTrials.gov registration NCT04666324) was to determine the effects of a nine-strain synbiotic and simethicone on the duration of crying and the gut inflammation marker calprotectin in colicky babies aged 3–6 weeks, diagnosed using the Wessel criteria. The open-label study comprised a control group of non-colicky babies (n = 20) and two parallel treatment groups (each n = 50) to which colicky babies were randomly and equally assigned to receive the multi-strain synbiotic or simethicone orally for 28 days. Primary outcome measures were the change in daily crying duration and the level of fecal calprotectin on days 1 and 28 of the study. Administration of the synbiotic resulted in a rechange of crying duration of −7.18 min/day of treatment, while simethicone had a significantly smaller effect (−5.74 min/day). Fecal calprotectin levels in colicky babies were significantly elevated compared to those in non-colicky babies. Treatment with the nine-strain synbiotic resulted in a significant lowering of fecal calprotectin at the end of the study, while no such effect was found for simethicone. No adverse effects were reported. Study results confirm earlier findings of crying duration reductions in colicky babies by the synbiotic, an effect that might be linked to its anti-inflammatory properties.

## 1. Introduction

Infants suffering from infantile colic exhibit crying, fussing, or irritability which are recurrent, long-lasting, occur without an obvious cause, and cannot be resolved or prevented by caregivers [1]. Pediatricians often employ the “rule of three” for identifying colicky babies. This rule is based on the Wessel criteria, which comprise periods of crying and restlessness for longer than three hours, occurring more than three days per week for more than three weeks [2]. In most cases, infantile colic is a self-limiting condition which will disappear three to five months after birth [3]. Occurrence rates found in publications vary from 3% to 40%, a variability that is most likely related to differences in the diagnostic criteria employed in the respective studies [4]. Under normal circumstances, infantile colic resolves over time without treatment. However, the stress caused by infantile colic for newborns, their parents, and healthcare providers makes it a leading reason why parents consult pediatricians in the early phase of infancy [5]. In addition, infantile colic is strongly associated with maternal depression [6] and early termination of breastfeeding [7]. While it is a rare event, abusive head trauma (AHT), including shaken baby syndrome (SBS), occurs more frequently in colicky babies than in normal infants [8].

The etiology of infantile colic remains unclear, but a disturbed gut microbiota might play a crucial role in understanding the cause [9,10]. Analyses of the gut microbiota of colicky babies reveal that bacterial colonization is slow and results in a less diverse and stable bacterial community [11]. With the cause of infantile colic remaining unclear, common treatment of infantile colic comprises a variety of approaches, among them manual therapy, administration of probiotics, and simethicone. A recently published systematic review found evidence that probiotics were effective for reducing crying time in infants. By contrast, evidence for significant effects of manual therapies and simethicone were only moderate or low, respectively [12]. Probably the best-characterized probiotic for the treatment of infantile colic is *Limosilactobacillus (L.) reuteri* DSM17938, however, reduction in crying duration caused by this probiotic has mainly been seen in exclusively breastfed infants [9,13,14,15]. Probiotic preparations containing *Lacticaseibacillus rhamnosus* GG were found to have no effect on crying duration [16,17]. Among the few studies evaluating the effects of multi-strain synbiotics in infantile colic, two have shown significant reductions in crying duration [18,19].

As a disturbed gut microbiota is often associated with gut inflammation, it has been suggested that inflammatory processes in the gut might play a critical role in infantile colic. In a recently published study, it was shown that the gut inflammation marker fecal calprotectin is elevated in colicky babies compared to non-colicky babies, and that administration of the probiotic *L. reuteri* DSM17938 resulted in the lowering of fecal calprotectin [20]. Calprotectin is a calcium- and zinc-binding protein mainly found within neutrophils. The presence of calprotectin in feces is a consequence of neutrophil migration into the gastrointestinal tissue due to inflammatory processes [21]. Fecal calprotectin concentration demonstrates good correlation with intestinal inflammation and is commonly used as a biomarker for gastrointestinal disorders [22]. In this respect, it is an advantage that calprotectin is stable in fecal samples at room temperature, which makes sample handling easy.

The present study was triggered by four reasons: (i) results from a survey among Polish and German pediatricians showing that pediatricians are using products containing probiotic microorganisms (among them the nine-strain synbiotic investigated in the present study) or simethicone for the treatment of infantile colic [23], (ii) the finding that the nine-strain synbiotic inhibits in vitro certain pathogenic bacteria associated with gut inflammation [24], (iii) a number of anecdotal reports from pediatricians that the nine-strain synbiotic reduces crying in colicky babies, and (iv) results from an earlier clinical trial showing that administration of the multi-strain synbiotic resulted in significantly stronger improvements of key crying measures (number of crying days, duration of crying, and number of crying events) in colicky babies when compared with simethicone [19]. In the study presented here, the effects of simethicone and the multi-strain synbiotic on the daily crying duration of colicky babies were assessed over a 28 day treatment period. In addition, we investigated the levels of fecal calprotectin in both treatment groups and in a control group of non-colicky babies before treatment (day 1) and at the end of treatment (day 28), aiming to collect information regarding the gut inflammatory situation in colicky babies and how it is influenced by the administration of simethicone or the nine-strain synbiotic.

## 2. Materials and Methods

The study was conducted at the GP Clinic “Pro Familia”, 62-028 Kozieglowy, Poland and GP Clinic “Panaceum” 27-230 Brody, Poland between December 2020 and September 2021. The trial was registered at ClinicalTrials.gov (NCT04666324).

Diagnosis of infantile colic based on the Wessel criteria was performed as part of the standard set of examinations in all newborns (aged 3–6 weeks) in both centers [2]. Parents of colicky babies and of a number of non-colicky babies were asked for their consent to participate in the study. Babies for whom consent was obtained were assessed for eligibility to participate in the study. Patients who were previously treated with probiotics, synbiotics, or antibiotics, and those with crying because of organic causes, were excluded from participation in the study. Data regarding type of delivery, weight at birth, gestational age, and feeding details were collected with a questionnaire completed by parents supported by midwives or nutritionists.

Non-colicky babies were assigned to a control group (arm 1, n = 20) for the measurement of calprotectin levels on the day of enrolment (day 1) and after four weeks (day 28). Colicky babies were equally randomized and assigned to either a group treated with simethicone (arm 2, n = 50) or to a group treated with the multi-strain synbiotic (arm 3, n = 50). Calculation of sample sizes needed for the study were based on experiences from our earlier clinical trial [19] and the usage of the Sample Size Calculator of Sealed Envelope Ltd. (London, UK). We assumed a responder rate of 50% in the simethicone group and a responder rate of 80% in the nine-strain synbiotic group. Other parameters used for the calculation were a significance level of 5% and a power of 50%. We calculated with a dropout rate of 0%, as in our earlier trial we had zero loss of enrolled patients. The Simple Randomiser of Sealed Envelope Ltd. (London, UK) was used to create a two-treatment equal allocation randomization scheme. This scheme was used by pediatricians to allocate patients to the two treatment arms of study on the basis of their entry into the study. Patients of study arm 2 were treated for four weeks, three to six times per day, with 6 drops of simethicone (Espumisan^®^, 100 mg/mL, Berlin Chemie/Menarini Polska sp. Z o.o., Warsaw, Poland). Patients of study arm 3 were treated for four weeks with one stick pack of a nine-strain synbiotic (Vivatlac^®^ Baby, Vivatrex GmbH, Rees, Germany) per day. Each stick pack of the multi-strain synbiotic contains a total of 10^9^ colony forming units (CFUs) with equal CFU amounts of the following probiotic bacteria: *Lactobacillus acidophilus* LA-14, *Lacticaseibacillus casei* R0215; *Lacticaseibacillus paracasei* Lpc-37; *Lactiplantibacillus plantarum* Lp-115; *Lacticaseibacillus rhamnosus* GG, *Ligilactobacillus salivarius* Ls-33, *Bifidobacterium (B.) lactis* Bl-04, *B. bifidum* R0071, *B. longum* R0175, and 1.43 g of the prebiotic fructooligosaccharides (FOS). Parents were provided with a diary (24-h Parental Daily Report) to record crying behavior, drug administration, and side effects during the 28 days of the study.

For the determination of fecal calprotectin concentrations, samples were taken from all patients in all three groups on day 1 and day 28. Collection of samples was performed during physician visits with the cooperation of parents assisted by a nurse or at home by the parents. In all cases samples were immediately frozen at −18 °C. Samples were transported by using special cooling containers. Stool samples were stored at −18 °C until further processing. Single-use Calex^®^ Caps (Bühlmann Laboratories, Schönenbuch, Switzerland) were used according to the manufacturer’s instructions to prepare samples for measurements. Calprotectin concentrations were determined using QB^®^fCAL extended test (Bühlmann Laboratories, Schönenbuch, Switzerland) in combination with a Quantum Blue^®^ Reader II BI-POCTR-ABS (Bühlmann Laboratories, Schönenbuch, Switzerland).

Normality of the analyzed variables was assessed visually using quantile–quantile plots. Statistical analyses were conducted in the R environment 2021 [25]. The differences between the three arms of the trial presented in Table 1 were assessed using the chi-squared test for the count data and the ANOVA test for continuous variables. The ANOVA test was followed by the Tukey post-hoc test corrected for multiple comparisons available in the rstatix package [26]. Differences between the calprotectin level on the first and last day of the trial were assessed using a regression model employing the following equation:(1)$$y_{ij}=\beta_{0}+\beta_{1}arm_{i}+\beta_{2}day_{j}+\beta_{3}arm_{i}day_{j}+e_{ij},$$
where *y* is the calprotectin level for *i*th trial arm (control, simethicone, or multi-strain synbiotic), *j*th trial day (first or last). The βs are regression coefficients, and ***e*** is a random error term.

Significance of the differences between the levels of analyzed effects was assessed based on pairwise comparison of the estimated marginal means using the emmeans package [27]. Differences between the speed in which the crying duration decreased over the consecutive days of the trial was assessed using a linear mixed-effect model employing the following equation:(2)$$y_{ijk}=\beta_{0}+\beta_{1}arm_{i}+\beta_{2}day_{j}+\beta_{3}arm_{i}day_{j}+{\mathrm{patient}}_{k}+e_{ijk},$$
where *y* is the calprotectin level for the *i*th trial arm (simethicone, or multi-strain synbiotic), *j*th trial day (first or last) for the *k*th patient. The *β*s are regression coefficients, and ***patient*** and ***e*** are random terms. This model was fitted using the lme4 package [28] and the significance of the difference between the slopes for the arms of the trial was assessed using the emmeans package [27].

## 3. Results

### 3.1. Patient Flow, Study Progress and Baseline Characteristics of Patient Groups

Figure 1 shows the flow of the patients through the study. During the recruitment period of the study, a total of 1248 babies were born in the two study centers. A total of 1122 (89.9%) were evaluated using the Wessel criteria for infantile colic. Of these, 242 (21.6%) newborns qualified as colicky babies; 117 were excluded because their parents declined to participate in the study, and 25 because the infants met one or several of the exclusion criteria. The remaining 100 newborns were randomly and equally allocated to the simethicone treatment group (arm 2) or the multi-strain synbiotic treatment group (arm 3). Twenty newborns of the 880 non-colicky babies were recruited to the control group (arm 1) of the study for the measurement of fecal calprotectin with the consent of their parents. Two patients in the treatment groups were excluded from the final analyses due to protocol violation (wrong allocation of treatment) and five patients were excluded because they received antibiotics during the treatment period. Table 1 shows the baseline characteristics of the three study groups.

Baseline characteristics of the control (arm 1), the simethicone (arm 2), and the multi-strain synbiotic groups (arm 3) are shown in Table 1. Statistical analyses revealed that there were no significant differences between the three groups with respect to gender, type of delivery, age at enrolment, or feeding. There were also no significant differences in the crying behavior measures of the patients allocated to the simethicone and the multi-strain group. In these two treatment groups the crying behavior was significantly different from that determined in the control group, with elevated levels of crying in all three assessed measures.

### 3.2. Effect of Simethicone and of the Multi-Strain Synbiotic on Crying Duration

Effects of the administration of simethicone or the multi-strain synbiotic on daily crying duration are shown in Figure 2. In both treatment groups, the duration of daily crying gradually decreased over the course of the trial. Linear regression analysis revealed that in the simethicone group, crying duration was changed by −5.74 min per day of treatment, while treatment with the multi-strain synbiotic resulted in a change by −7.18 min per day of treatment. The difference between the slopes was 1.44 min/day of treatment, which was found to be a highly significant difference. As revealed by the regression analysis of the daily data, average daily crying duration at the end of treatment (day 28) was 75.3 min in patients of the simethicone group and 36.1 min in those of the multi-strain synbiotic group.

No adverse effects were reported for patients in any of the study arms.

### 3.3. Fecal Calprotectin

Results of the fecal calprotectin measurements are shown in Figure 3. The fecal calprotectin levels in non-colicky babies (control group) were similar on day 1 and on day 28. In colicky babies, the calprotectin levels on day 1 were significantly higher, both in the simethicone group and the multi-strain synbiotic group, compared to those in non-colicky babies. There was no significant difference between the fecal calprotectin levels of the simethicone and multi-strain groups on day 1 of the study. On day 28, fecal calprotectin levels in the simethicone groups were slightly reduced, but this reduction was not significant when compared with the level on day 1. In contrast, treatment with the multi-strain synbiotic resulted in a significant lowering of the fecal calprotectin levels on day 28. However, the calprotectin levels in the multi-strain synbiotic group on day 28 were still significantly higher than those in the control group on day 28.

## 4. Discussion

Although infantile colic is generally a self-resolving condition, its stressful character for caregivers and its association with certain risks (maternal depression, early weaning, shaken baby syndrome) urges pediatricians to intervene by prescribing. There is little evidence that the effects of simethicone in colicky babies is different from those of placebo administration [5,29]. Nevertheless, simethicone is still widely employed by pediatricians (including in the centers participating in the present study) for the pharmacological treatment of colicky babies [23,30]. By contrast, the administration of certain products containing probiotic bacteria has been demonstrated to significantly reduce crying duration compared to the administration of a placebo or simethicone [9,13,14,15,18]. The results of the present study confirm the findings of a previous study where a nine-strain synbiotic resulted in a significant reduction of the crying duration in colicky babies when compared with the effects of simethicone [19]. At the end of treatment (day 28) in the present study, the difference between the average daily crying duration in the simethicone and the multi-strain synbiotic group was 39.2 min—a difference that is not only statistically significant but is also most likely appreciated by parents stressed by their colicky babies. Analysis of the effect of the multi-strain synbiotic on the crying duration after excluding formula-fed and mixed-fed newborns from the analysis (data not shown) revealed that there was no significant difference in the effect size in this subgroup of newborns. This property of the multi-strain synbiotic might translate into a clinical advantage over *L. reuteri* DSM 17938, which has been demonstrated to be particularly effective in exclusively breastfed newborns [12]. However, such a claim would require the performance of a clinical study comparing effects of the multi-strain synbiotic head-to head with those of *L. reuteri* DSM 17938 in colicky babies, a task that we reserve for a future clinical trial.

The gut microbiota of colicky babies differs significantly from that of non-colicky controls, specifically in terms of (i) a slower colonization with lower diversity and stability, (ii) less butyrate-producing bacteria, (iii) a high abundance of Proteobacteria (including species producing gas and inflammation) and (iv) lower amounts of lactobacilli and bifidobacteria (including species with anti-inflammatory effects) [11]. The presence of inflammation in the gut of colicky babies has been suggested by the finding of elevated levels of fecal calprotectin [20], which is an established marker of gut inflammation [22]. Supporting this, in the present study, the fecal calprotectin levels in colicky babies were significantly elevated compared to those found in non-colicky babies, indicating that gut inflammation might play a role in the etiology of infantile colic. The mixture of probiotic strains contained in the multi-strain synbiotic investigated in the study comprises lactobacilli and bifidobacteria, which have been demonstrated to inhibit—both individually and as a mixture—the in vitro growth of certain pathogenic bacteria. This property could have anti-inflammatory effects [24]. The finding that the administration of the multi-strain synbiotic significantly lowered fecal calprotectin levels in colicky infants supports this. More studies will be needed to investigate the gut inflammation in colicky babies, its causes, and by what mechanisms the administration of the multi-strain synbiotic suppresses this inflammation. So far, a calprotectin-lowering effect in colicky babies has only been described for the treatment of colicky babies with *L. reuteri* DSM17938 [20]. As found in the present study, simethicone does not lower fecal calprotectin levels, which might explain the lack of effects in colicky babies when compared with those of a placebo.

This study and the interpretation of its results have certain limitations. Our initial sample size calculation was based on the (in retrospect) optimistic assumption of a dropout rate of zero. In reality, we lost 7 patients for different reasons, resulting in the fact that data from only 93 patients (46 in the simethicone treatment group and 47 in the nine-strain synbiotic treatment group) could be analyzed. Despite the small number of patients included in the study and the higher-than-expected dropout rate, measured effects reached statistical significance. The administration of 6 drops of a 100 mg/mL solution of simethicone 3–6 times a day represents common practice in the treatment of colicky babies, which we aimed to replicate in our study. Therefore, the amount of simethicone administered to each individual cannot be exactly stated. Assessing crying behavior by the use of parental diaries has clear limitations. These limitations have been reviewed by several authors [31,32]. Nevertheless, and with the current lack of instruments to determine crying durations in infants more accurately, it has been shown that paper diary recordings can provide reasonably good estimates of crying durations [33]. Another major limitation of the study is that medication was not blinded—a fact that is attributed to the fact that we investigated commercial products for which respective placebos were not available for our research, and budget restrictions did not allow us to produce them.

## 5. Conclusions

Results from the present study indicate that the multi-strain synbiotic can be considered as a treatment option aiming to reduce the crying duration in colicky babies. The positive effects on the duration of crying might be linked to an anti-inflammatory property of the multi-strain synbiotic, as it was found to reduce fecal calprotectin levels, an established marker of gut inflammation. By contrast, the administration of simethicone resulted in a lesser reduction of crying duration and had no effect on fecal calprotectin. Data from the study also suggest that the measurement of fecal calprotectin can be employed for supporting the diagnosis of infantile colic as well as the monitoring of treatment effects. As fecal calprotectin levels were still significantly higher than those in non-colicky babies even after four weeks of treatment with the multi-strain, longer treatment durations should be investigated in future studies.

## Figures and Tables

**Figure 1 microorganisms-10-00430-f001:**
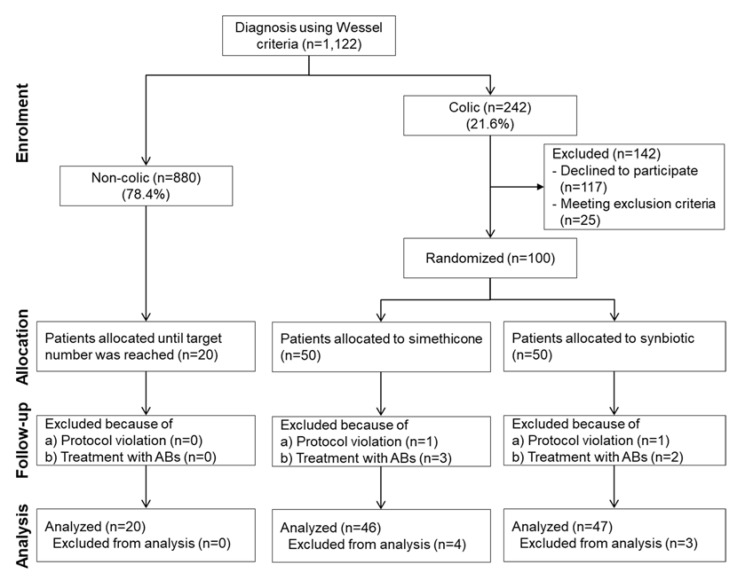
Patient enrollment and study progress.

**Figure 2 microorganisms-10-00430-f002:**
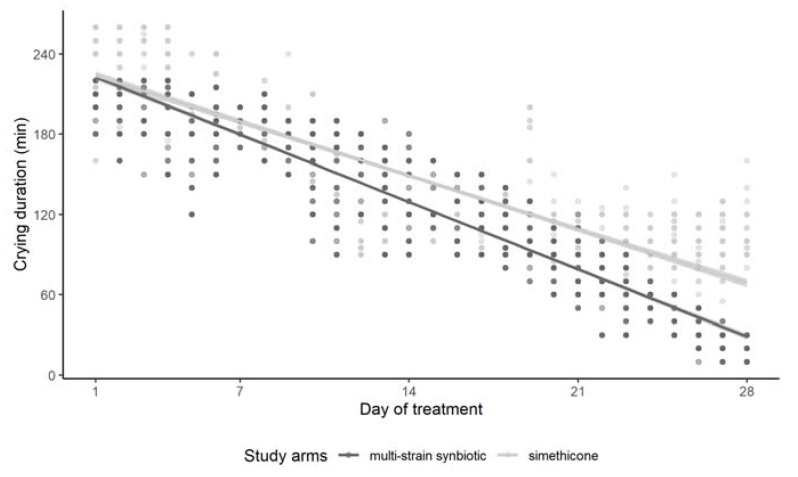
Effects of simethicone or multi-strain synbiotic treatment in colicky babies on daily crying duration over the course of the treatment.

**Figure 3 microorganisms-10-00430-f003:**
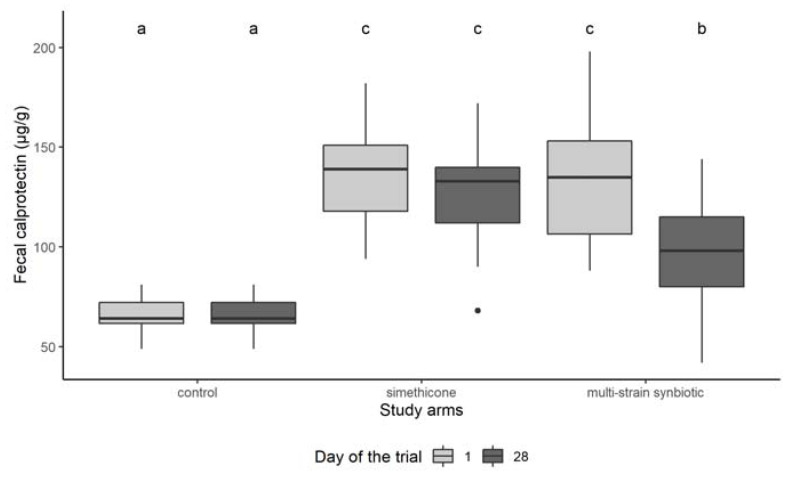
Fecal calprotectin levels in the control, simethicone, and multi-strain synbiotic groups on day 1 (before treatment) and on day 28 (at end of the treatment period) of the study. Different lowercase letters indicate significant differences between groups (*p* < 0.05).

**Table 1 microorganisms-10-00430-t001:** Baseline characteristics of control, simethicone, and multi-strain synbiotic groups.

		Control (n = 20)	Simethicone (n = 46)	Multi-Strain Synbiotic (n = 47)
**Basic** **Characteristics**	Gender ^1^ (female/male)	9/11 ^a^	18/28 ^a^	24/23 ^a^
	Delivery ^1^ (normal/caesarean)	15/5 ^a^	28/18 ^a^	29/18 ^a^
	Feeding ^1^ (breast/formula/mixed)	13/5/2 ^a^	25/17/4 ^a^	32/10/5 ^a^
	Birthtime in pregnancy ^2^ (week)	39.00 ^a^ ± 2.83 ^a^	39.48 ^a^ ± 2.21 ^a^	39.85 ^a^ ± 2.11 ^a^
	Birthweight ^2^ (g)	3365.0 ^a^ ± 417.4	3383.5 ^a^ ± 289.0	3501.9 ^a^ ± 372.0
	Age at enrollment ^2^ (days)	33.15 ^a^ ± 4.44	32.41 ^a^ ± 5.48	31.41 ^a^ ± 6.46
**Crying** **Behavior**	Crying days * last 3 weeks before treatment ^2^	NA	14.3 ^a^ ± 1.19	14.5 ^a^ ± 2.18
	Avg. crying duration (min/day) last 3 weeks before treatment ^2^	93.10 ^A^ ± 25.85	218.04 ^B^ ± 20.51	221.70 ^B^ ± 24.08
	Avg. crying phases/day last 3 weeks before treatment ^2^	4.1 ^A^ ± 0.85	5.98 ^B^ ± 0.95	5.89 ^B^ ± 1.07

^1^ Significance of differences between count data assessed with chi-squared data. ^2^ Significance of differences assessed with ANOVA followed by Tukey post-hoc test corrected for multiple comparisons. Level of significance of differences indicated with letters: same letter = no significant difference, small letters = *p* < 0.05, capital letters = *p* < 0.01. * Crying day defined as a day with crying duration ≥ 3 h.

## Data Availability

The data presented in this publication are openly available in FigShare at https://doi.org/10.6084/m9.figshare.18627203 (accessed on 18 January 2022).

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
