# Peer review of "Nine-Strain Bacterial Synbiotic Improves Crying and Lowers Fecal Calprotectin in Colicky Babies—An Open-Label Randomized Study"

_microorganisms, 2022, doi:10.3390/microorganisms10020430_

Round 1

Reviewer 1 Report

The aim of the study is clearly stated both in the abstract and at the end of the introduction.

The most recent taxonomy of lactobacilli is used correctly.

The experimental design is adequate. Nevertheless, the study has some limitations, which are however assumed by the authors.

The authors refer that a calprotectin lowering effect in colicky babies had been described before for the treatment of colicky babies with L. reuteri DSM17938. What is the advantage of using this multi-strain synbiotic instead of the single L. reuteri strain?

Elevated levels of fecal calprotectin indicate gut inflammation, but can the authors specifically correlate these levels with particular genera/species within the gut microbioma? Of all 3 groups? This would be very interesting...

Reviewer 2 Report

The topic of the study is of interest, and it is well presented. However, I consider that authors should consider the following comments:

  1. When calculating sample size, ¿why authors consider as cero the dropout? In my opinion, and based in literature, a minimum dropout of 10% would be expected.
  2. Flow diagram show that, in fact, the final number of participants (those that were analyzed) was lower than the one calculated. Even considering that a significant effect was found (with a lower sample size than the one calculated), this fact should be added as a limitation of the study.
  3. In the text it is stated that simethicone was administered 3-6 times a day. Therefore, some newborns could have ingested 3 times/day, some others 4, some others 5 and some others 6 times/day. Would have this affected the internal validity of the study? In any case, this should be added as a study limitation.
  4. How were fecal samples obtained? At home by the parents? Kept at the home freezer? If so, how were samples sent to the place of analysis? A more detailed description regarding this process is needed.
  5. Was the normality of the data assessed? Please, confirm in the text.
  6. Figure 1: In my opinion is not necessary to add percentages.
  7. Please, explain in discussion why the synbiotic used was expected to reduce infant colic. A more detail explanation regarding possible mechanisms involved is needed.
  8. Regarding the effect of the interventions on the mean crying time per day, results are statistically significant. However, authors should discuss the clinical significance of this result. Regarding crying time/day, to be consider a colic, babies must cry for more than 3 hours. Do authors think that a 7.18-minute reduction would make a clinical difference even if statistically significant? This should be explained in discussion.
  9. Following with the previous comment, synbiotic treatment reduced crying time compared to simethicone. However, the difference, although statistically significant, was only of around 2 minutes. Authors should include in discussion which other advantages implies the use of the synbiotic compared to the use of simethicone other than the crying time reduction (which clinically does not make much difference).

Reviewer 3 Report

The manuscript "Nine-strain bacterial synbiotic improves crying and lowers fecal calprotectin in colicky babies - an open-label randomized study" by Bernatek et al attempts to determine the effect of the synbiotic on reducing colick compared to simethicone, an anti-foaming agent used to reduce gas, specifically by reducing levels of calprotectin, which is a biomarker of intestinal inflammation.  The current study is an expansion on a recently published clinical study by the same authors, to increase the study duration and also look at calpectin levels.

Overall, the study does show lowering of calpectin by the synbiotic and maybe a reduction in crying duration (conflicting information between texts and figures make this unclear as detailed below).  However, the study is lacking in providing rationale on why the specific synbiotic was chosen, details on synbiotic administration, tracking colonization of synbiotic or its impact on the gut microbiota.  While the last point is valuable but not critical, the other points are critical for readers to understand.

Introduction

It is unclear why the 9 probiotic strains were chosen for the study.  Authors spend some time discussing the beneficial impact of L. reuteri on infantile colic, which wasn't included in the probiotic.  They also discuss the lack of a benefit of L. rhamnosus, which was included in the probiotic.  No other information is given about the other 8 strains on why they were included in the synbiotic.  These are important details that are missing.  Was the synbiotic a commercial formulation that has shown efficacy for intestinal inflammation in other studies?  Was it developed by the authors?  Are there previous studies that give us a clue about what the bacteria are doing?  Could the synbiotic be reduced down to a single species?  There is some vague mention in the discussion that the strains included inhibits the growth of some pathogenic bacteria which could maybe have anti-inflammatory effects.  The rationale behind synbiotic choice, including all strains involved should be up-front.

While the authors describe the duration of administration for simethicone (4 weeks), no such details are provided for the synbiotic.  How long were the synbiotics taken?  Additionally, did the synbiotics colonize or are their effects transient such that parents would have to administer the synbiotics until colic resolves on its own?  No attempts were made to track colonization either through culture-based or molecular means nor were any attempts made to see if the synbiotic acted by altering microbiota composition as is implied later in the discussion.

One of the main conclusions is a significantly greater reduction in daily crying duration between the synbiotic and simethicone by less than 1.5 minutes.  In total, the drop in crying was 5.74 minutes for the synbiotic and 7.18 minutes for simethicone.  These data were quantified by the parent's diary for crying behavior.  While the numbers may be significant, I have to question the actual impact on the daily lives of the infants/parents.  I know from being a parent of 2 young children, that accurately detailing crying duration 24hrs a day, 7 days a week, especially for new parents would be impossible, especially since they would need accuracy to the second for 1.5 minutes to be meaningful.  

Additionally, these reported numbers seem to be in conflict with Figure 2 which shows a drop from ~230 minutes of crying/day to 50-80 minutes (150-180 minutes total depending on treatment group).  These results would be much more behaviorally significant in addition to statistically significant.  Please clarify the difference between the figure and the text.

Round 2

Reviewer 3 Report

The authors have sufficiently revised their manuscript addressing the major critiques that I previously had.